# Exploring the Validity of the 14-Item Mediterranean Diet Adherence Screener (MEDAS): A Cross-National Study in Seven European Countries around the Mediterranean Region

**DOI:** 10.3390/nu12102960

**Published:** 2020-09-27

**Authors:** María-Teresa García-Conesa, Elena Philippou, Christos Pafilas, Marika Massaro, Stefano Quarta, Vanda Andrade, Rui Jorge, Mihail Chervenkov, Teodora Ivanova, Dessislava Dimitrova, Viktorija Maksimova, Katarina Smilkov, Darinka Gjorgieva Ackova, Lence Miloseva, Tatjana Ruskovska, Georgia Eirini Deligiannidou, Christos A. Kontogiorgis, Paula Pinto

**Affiliations:** 1Research Group on Quality, Safety and Bioactivity of Plant Foods, Campus de Espinardo, Centro de Edafologia y Biologia Aplicada del Segura-Consejo Superior de Investigaciones Científicas (CEBAS-CSIC), P.O. Box 164, 30100 Murcia, Spain; mtconesa@cebas.csic.es; 2Department of Life and Health Sciences, University of Nicosia, 1700 Nicosia, Cyprus; Philippou.e@unic.ac.cy (E.P.); chris_paf@hotmail.com (C.P.); 3National Research Council (CNR) Institute of Clinical Physiology, 73100 Lecce, Italy; marika@ifc.cnr.it; 4Laboratory of Biochemistry and Molecular Biology, Department of Biological and Environmental Sciences and Technologies, University of Salento, 73100 Lecce, Italy; stefanoquarta2@gmail.com; 5Polytechnic Institute of Santarém, School of Agriculture, 2001-904 Santarém, Portugal; vanda.andrade@esa.ipsantarem.pt (V.A.); rui.jorge@esa.ipsantarem.pt (R.J.); 6Life Quality Research Centre (CIEQV), IPSantarém/IPLeiria, 2040-413 Rio Maior, Portugal; 7Centro de Investigação Interdisciplinar Egas Moniz (CiiEM), Instituto Universitário Egas Moniz, 2829-511 Monte de Caparica, Portugal; 8Slow Food in Bulgaria, 9 Pierre De Geytre St. bl. 3, 1113 Sofia, Bulgaria; vdmchervenkov@abv.bg (M.C.); tai@bio.bas.bg (T.I.); dessidim3010@gmail.com (D.D.); 9Faculty of Veterinary Medicine, University of Forestry, 1797 Sofia, Bulgaria; 10Department of Plant and Fungal Diversity and Resources, Institute of Biodiversity and Ecosystem Research, Bulgarian Academy of Sciences, 1113 Sofia, Bulgaria; 11Faculty of Medical Sciences, University Goce Delcev, str. Krste Misirkov, No. 10-A, POB 201, 2000 Stip, North Macedonia; viktorija.maksimova@ugd.edu.mk (V.M.); katarina.smilkov@ugd.edu.mk (K.S.); darinka.gorgieva@ugd.edu.mk (D.G.A.); lence.miloseva@ugd.edu.mk (L.M.); tatjana.ruskovska@ugd.edu.mk (T.R.); 12Laboratory of Hygiene and Environmental Protection, School of Medicine, Democritus University of Thrace, Dragana, 68100 Alexandroupolis, Greece; edeligia@med.duth.gr (G.E.D.); ckontogi@med.duth.gr (C.A.K.)

**Keywords:** Mediterranean Diet, validation, dietary health benefits, dietary assessment, food frequency questionnaire, body mass index, diet quality, diet adherence, survey instruments, Southern Europe, dietary assessment

## Abstract

This study provides comprehensive validation of the 14-item Mediterranean Diet Adherence Screener (14-MEDAS) in an adult population from Greece (GR), Portugal (PT), Italy (IT), Spain (SP), Cyprus (CY), Republic of North Macedonia (NMK), and Bulgaria (BG). A moderate association between the 14-MEDAS and the reference food diary was estimated for the entire population (Pearson *r* = 0.573, *p*-value < 0.001; Intraclass Correlation Coefficient (ICC) = 0.692, *p*-value < 0.001) with the strongest correlation found in GR, followed by PT, IT, SP, and CY. These results were supported by kappa statistics in GR, PT, IT, and SP with ≥50% of food items exhibiting a fair or better agreement. Bland–Altman analyses showed an overestimation of the 14-MEDAS score in the whole population (0.79 ± 1.81, 95%Confidence Interval (CI) 0.61, 0.96), but this value was variable across countries, with GR, NMK, and BG exhibiting the lowest bias. Taking all analyses together, the validation achieved slightly better results in the Mediterranean countries but a definitive validation ranking order was not evident. Considering growing evidence of the shift from Mediterranean Diet (MD) adherence and of the importance of culture in making food choices it is crucial that we further improve validation protocols with specific applications to measure and compare MD adherence across countries and to relate it to the health status of a specific population.

## 1. Introduction

The dietary pattern associated with the Mediterranean Diet (MD) is characterized by the daily consumption of olive oil (mainly extra-virgin or virgin olive oil) as the main source of fat, whole grains, fruits, and vegetables, weekly consumption of legumes, nuts, fish, and wine in moderation, as well as a moderate intake of lean fresh meat, and dairy products. This diet provides an important source of minerals, vitamins, mono- and polyunsaturated fatty acids, fiber as well as a broad range of bioactive antioxidant and anti-inflammatory compounds [1]. The MD is promoted worldwide as one of the healthiest dietary patterns due to its consistently attributed benefits against chronic diseases and its association with longevity [2,3]. An umbrella review of meta-analyses of observational studies and randomized clinical trials including a total population of over than 12,800,000 subjects established that a greater adherence to the MD reduced the risk of overall mortality, cardiovascular diseases, cancer incidence, neurodegenerative diseases as well as type 2 diabetes [4].

A critical issue to further progress in the understanding of the impact of the MD on health is the implementation of suitable tools that allow for the assessment of MD adherence. Various models of food diaries, food frequency questionnaires and dietary assessment surveys have been developed, validated, and tested in different populations and countries [5]. Particularly, the PREDIMED study, a primary prevention nutrition-intervention trial, led to the development of the “Mediterranean Diet Adherence Screener” (MEDAS) to assess dietary intake [6]. This instrument comprises of 14 questions regarding the main groups of food consumed as part of the MD and was validated against a 136-item food frequency questionnaire (FFQ). The 14-item MEDAS questionnaire was indicated to be a moderate and reasonably valid tool for the rapid estimation of MD adherence. In addition, a higher MEDAS score was positively associated with high-density lipoprotein (HDL) cholesterol, and negatively associated with body mass index (BMI), waist circumference, triglycerides (TG), fasting glucose, and total cholesterol/HDL ratio showing its potential applicability in clinical practice. A limitation attributed to this study was that the findings could not be extrapolated to the general population since the participants were constituted by elderly individuals at a high risk for coronary heart disease (CHD) [6].

The 14-MEDAS has been subsequently investigated as a tool to assess MD adherence in various other countries, i.e., Germany [7], USA [8], UK [9], or Korea [10]. Although validation was conducted using FFQs [7,8,10] or food diaries [9] with different designs, all those studies also reported an overall modest-to-fair value of concordance with the 14-MEDAS and corroborated that this instrument constituted a moderately valid tool to assess MD adherence in their respective countries. The 14-item MEDAS was also translated into a Persian language and validated in the north of Iran where a 13-item modified version was found to be reliable and sufficiently valid in a high-risk population [11]. These studies were all conducted in non-Mediterranean countries, and some were specifically carried out in subpopulations with high body mass index (BMI) and (or) high risk of cardiovascular disease [6,9,10]. Regarding the Mediterranean region, a modified 15-item questionnaire (QueMD) was developed to measure the adherence to MD in Italy and reported to exhibit moderate correlation with the reference FFQ and poor-to-fair agreement depending on the food item. Nevertheless, it was regarded as a useful tool to assess adherence to the MD in the Italian population [12]. Moreover, a 17-item adapted Israeli MD screener was proven to be reliable and of predictive utility for mortality in an adult Israeli population [13] and, more recently, a Portuguese version of the 14-MEDAS was validated against a FFQ with an overall fair agreement [14].

Although the validity of the MEDAS tool was examined in all those individual countries [7,8,9,10,11,12,13,14], a simultaneous and comparative cross-national validation of the 14-MEDAS tool in different countries of the Mediterranean region has never been undertaken. This is important due to recent considerable changes in the adherence to MD in several Mediterranean countries [15,16] and growing evidence of the importance of culture in making “healthy” food choices [17,18]. The main aim of this research was to explore the applicability of a 14-item MEDAS questionnaire to the general adult population from several Southern European countries around the Mediterranean area with different dietary cultures, i.e., Greece (GR), Cyprus (CY), Italy (IT), Spain (SP), and Portugal (PT), as well as to two adjacent Balkan countries, Republic of North Macedonia (NMK) and Bulgaria (BG), to further support the validity of this tool as a rapid and flexible instrument to measure and differentiate adherence to the MD.

## 2. Materials and Methods

### 2.1. Study Design and Recruitment

Eligible participants were adults (age ≥ 18 years) of both sexes recruited through institutional and personal contacts of each of the study’s researchers as well as through social media. No gender, educational, health, social, and cultural selection were intended. Participants were asked to report their sex, age, as well as their weight and height, which were used to calculate their BMI (BMI = kg/m^2^). Based on an anticipated Intraclass Correlation Coefficient (ICC) of 0.5 (moderate agreement) and an accepTable 95% confidence interval (95%CI) width of 0.4 [19], as well as a minimum desirable size for Bland Altman plots [20], we aimed at a sample population of N~50 respondents per country. Responses were collected from June to December 2019.

The study protocol was approved by the Ethics Committee of each partner Research Institution in line with the principal approval obtained by the coordinating Institution (Polytechnic Institute of Santarém, Research Unit, Ref.: 022019 Agrária, Portugal). The study was performed according to the principles established by the Declaration of Helsinki. All participants signed an informed consent before enrollment. Data from food diaries and food frequency questionnaires were handled as personal data according to the European law and subjected to pseudonymisation before statistical analysis (a code was assigned to each participant’s food diary and food frequency questionnaire, by a different investigator from the one performing the analysis).

### 2.2. Validation Protocol

The validation process was carried out against a self-reported 3-day food diary (3d-FD). All participants were instructed by a member of the research team on how to fill the 3d-FD: (1) participants should not change their usual food consumption; (2) participants should record everything they ate and drank, including all snacks and beverages; (3) the record should be completed immediately after food or beverage intake and include the meal (breakfast, lunch, etc.), the name of each consumed food, information concerning the cooking method (boiled, roasted, stewed, fried in butter, olive oil, or cooking oil), and the amounts of each food in household measures (e.g., cups, teaspoons, tablespoons, etc., defined by each researcher, according to the country’s food habits). The participants had one week to fill in the food diaries (two non-consecutive weekdays and one weekend day), which were then collected by the researcher. The amounts indicated in the food diaries were converted to grams (g) of food using a published Portuguese photographic manual [21] and then to number of servings according to the servings defined in the 14-MEDAS.

Using the validated 14-MEDAS scoring instrument from the PREDIMED study [6], we designed a FFQ questionnaire to calculate the MEDAS score (FFQ-MEDAS) in the different countries taking part in the study (Appendix A). The questionnaire was thoroughly revised, discussed and then translated into the language of each participant country by a local committee formed by researchers involved in the study. A week after collection of the completed 3d-FD, a first FFQ-MEDAS (1) questionnaire was sent to the participants by the researcher via an e-mail link or presented via a personal interview. A second questionnaire FFQ-MEDAS (2) was sent or presented a week after collection of the first questionnaire. The criteria defined for the MEDAS (Appendix A) were used to calculate the Mediterranean diet adherence score from the 3d-FD and from the two separate questionnaires. The final MEDAS score can range between 0 and 14. For categorization of the adherence to the MD we applied the following criteria: weak adherence, ≤5; moderate to fair adherence, 6–9; good or very good adherence ≥10 [22].

### 2.3. Statistical Methods

Normality of the investigated variables in the different population samples was based on visual inspection of the frequency histograms and the Q-Q plots, the values of asymmetry and kurtosis, and sample size (*n* > 30) [23]. Quantitative normal variables were described as mean ± SD and categorical variables as relative frequency. The FFQ-MEDAS scores differences between countries, sexes, age ranges, and BMI categories were performed by one-way ANOVA followed by a post hoc analysis using the Tukey test.

The test–retest reliability (association between FFQ-MEDAS1 and FFQ-MEDAS2 responses) and the relationship between variables were estimated using the Pearson correlation coefficient (*r*). A correlation coefficient > 0.70 was considered a strong correlation; from 0.4 to 0.7 a moderate correlation; and <0.4 a weak correlation [24]. We also assessed the absolute agreement between the 3d-FD and the FFQ-MEDAS (mean score) by means of the ICC with their 95%CI and using the two-way mixed-effects model with average measures. The ICC values were considered as poor (<0.50), moderate (≥0.5 to <0.75), good (≥0.75 to 0.9), and excellent (>0.9) [25]. The absolute agreement between the responses that were obtained for each of the 14 food items derived from the FFQ-MEDAS and the corresponding ones from the 3d-FD was determined using κ statistics with their 95%CI. κ was computed for each of the two 14-item FFQ then averaged to provide a single value. For validation, the Cohen’s κ values were interpreted as κ ≤ 0 no agreement, κ = 0.01–0.20 slight agreement, κ = 0.21–0.40 fair agreement, κ = 0.41–0.60 moderate agreement, κ = 0.61–0.80 substantial agreement, κ = 0.81–1.0 almost perfect agreement. A negative kappa represents agreement worse than expected or disagreement with low negative values (0.00 to −0.10) generally being interpreted as “no agreement” and large negative kappa representing a great disagreement [26].

The agreement between the two methods was additionally verified by implementing the graphical analysis recommended by Bland and Altman which plots the “difference” in the score between the two methods (FFQ-MEDAS—3d-FD) on the *Y*-axis against the “mean value” of the two scores ((FFQ-MEDAS + 3d-FD)/2) on the *X*-axis [27]. Following recently specified criteria [28], we (i) report that the distribution of the differences was approximately normal and (ii) include complete Bland–Altman plots displaying lines for the mean difference (bias) and the Limits of Agreement (LOAs), as well as their corresponding 95%CI, (iii) numerical values, confidence intervals and standard deviation for mean differences, and (iv) numerical values and confidence intervals for LOAs. A linear regression analysis with the “difference” as the dependent value was used to indicate the direction of bias and whether it was constant across mean scores. All *p*-values were based on two-sided tests (bilateral significance), and those *p*-values < 0.05 were considered statistically significant. All analyses were conducted using SPSS version 26.0 statistical software (SPSS version 23.0; IBM Corp., Armonk, NY, USA).

## 3. Results

### 3.1. Characteristics of the Participants

From the initial total 414 voluntaries recruited, 402 participants completed the 3-d FD and the two 14-item questionnaires. The distribution and characteristics of the entire sample population and per country is displayed in Table 1. Overall, there was a higher participation of women than men (~60:40), but the distribution of sexes did not differ between countries (*p*-value = 0.775). The age of the participants ranged from 18 years old up to a maximum of 81 years old, with the main part of the sample population constituted by young and middle-aged adults (mean age = 39.4 ± 15.2 years). Younger participants prevailed in PT, CY, GR, and IT against a slightly older population in SP, NMK and BG (*p*-value < 0.001).

Regarding body weight, the mean BMI from the entire examined population was 25.2 ± 5.0 kg/m^2^ (slightly overweight), nevertheless, the largest proportion of participants (55.5%) was classified into the normal weight category. Moreover, the sample populations from SP, IT, PT, CY, and GR were mostly classified into the normal weight class. On the other hand, the sample population of NMK and BG included more than 50% of participants with overweight and obesity.

### 3.2. Validation of the FFQ-MEDAS against 3d-FD in the Selected Countries: Reliability, Correlation, and Agreement

Table 2 shows the results of the estimated FFQ-MEDAS and 3d-FD scores as well as the results of the validation analyses, both globally (all countries together) and per individual country. The reliability of the questionnaire (test–retest analysis) was good in all of the examined countries as shown by strong and very significant positive correlations between the FFQ-MEDAS (1) and the FFQ-MEDAS (2) (*r* > 0.8, *p*-value < 0.001), except for CY, which was slightly lower (*r* = 0.62; *p*-value < 0.001). The relationship between the FFQ-MEDAS mean score and the 3d-FD score for the entire population (i.e., all participating countries) exhibited a moderate positive correlation (*r* = 0.573, *p*-value < 0.001) with the strongest association found in GR (*r* = 0.895, *p*-value < 0.001), whereas SP, IT, PT and CY exhibited moderate significant positive correlation. The weakest significant association was found in BG (0.311, *p*-value = 0.016) while no correlation was detected in the case of NMK. The agreement between the FFQ-MEDAS and the 3d-FD was also estimated by the ICC index. Considering all the countries together, the average ICC displayed a moderate but significant agreement (0.692, 95%CI: 0.552, 0.780; *p*-value < 0.001). The best agreement was again found for GR with a maximum ICC value (0.939, 95%IC 0.887, 0.967; *p*-value < 0.001). PT, CY, and IT were all in the moderate range with ICC values above 0.5. A little poorer but still significant positive agreement was found in the case of SP (0.440, 95%CI −0.210, 0.745; *p*-value < 0.001) and BG (0.473, 95%CI 0.118, 0.686; *p*-value = 0.008) whereas there was no agreement in the case of the NMK.

Regarding the estimated MEDAS scores, the global FFQ-MEDAS value for all the Southern European countries examined was 6.22 ± 2.01 indicating a moderate-to-fair adherence to the MD. We did not find evidence for differences between sexes (men, 6.18 ± 2.01; women, 6.23 ± 2.01; *p*-value = 0.782) or in the correlation between FFQ-MEDAS and 3d-FD (*r* = 0.527, *p*-value < 0.001 and *r* = 0.609, *p*-value < 0.001 for men and women, respectively). There were no significant differences either in the estimated 14-MEDAS scores between the different age ranges (18–24 years, 6.25 ± 1.73; 25–34 years, 6.09 ± 1.97; 35–44 years, 6.16 ± 2.03; 45–54 years, 6.72 ± 2.18; 55–64 years, 6.14 ± 2.24; 65–81 years, 5.89 ± 2.09; *p*-value = 0.418) although we detected a small and almost significant negative correlation between age and 14-MEDAS (*r* = −0.1, *p*-value = 0.05). There were, however, significant differences between the countries (*p*-value < 0.001) with the highest score attained in the Spanish sample population (8.35 ± 1.65) whereas the lowest scores were found in NMK and BG (4.80 ± 1.65 and 4.47 ± 1.19, respectively; weak adherence). IT, PT, CY, and GR constituted a group with similar moderate-to-fair scores. On the other hand, the scores of the participants with overweight (5.83 ± 2.00) and obesity (5.67 ± 1.90) were significantly lower than those of the volunteers with normal weight (6.61 ± 1.98; *p*-value < 0.01). Our analyses also confirm a significant and negative weak correlation between the 14-MEDAS score and the BMI (*r* = −0.17, *p*-value = 0.001). We additionally found a significant and negative correlation between BMI and the variable “difference” between the 14-MEDAS and the 3d-FD scores (*r* = −0.13, *p*-value = 0.01).

### 3.3. Kappa Statistics: Analysis Per Food-Item and Country

Item-per-item agreement between the FFQ-MEDAS and the 3d-FD in the global sample population and in each separate country are detailed in Appendix A. These tables include the proportion of participants who met the criteria for achieving each of the 14 points of the score corresponding to each food item in the 3d-FD (prevalence) and in the 14-items FFQ, as well as the percentages of absolute and relative (κ statistics) agreement. Overall, a higher proportion of participants achieved the score 1.0 via the 14-MEDAS FFQ compared to the 3d-FD both globally and in each separate country. The highest difference was found in SP (~11%) and the lowest difference in GR and NMK (0.6–0.7%). On average (taking all food items together), the percentage of absolute agreement in the entire population was 72.3 ± 10.8% with the highest % of agreement reached for wine consumption (89.5%) and the lowest for vegetables (57.0%). Individually, most countries also reached a similar % of absolute agreement with GR exhibiting the highest overall agreement (81.2 ± 10.7%) and CY the lowest value (64.2 ± 14.9%).

Table 3 shows a comparative summary of the κ statistics results. Overall, a fair to moderate relative agreement was found for the whole population examined with the highest concordance detected for the consumption of olive oil (κ = 0.590) and fruits (κ = 0.502) and the poorest for vegetables (κ = 0.184) and red meat (κ = 0.114). Regarding individual countries, the best item-per-item concordance was found in GR, which exhibited most food items with a moderate agreement including three foods, i.e., nuts, white meat and “sofrito”, with a substantial or almost perfect agreement. SP, PT, and IT had all ≥ 50% of food items with a fair or higher level of agreement. On the other hand, CY, NMK, and BG had less than 30% of food items with a fair or better agreement. It should be noted though that while CY and NMK had the highest proportion of no-agreements or disagreements results, in the case of the sample population from BG, there was a high proportion of items (8 out of 14) that were not adequate for kappa analysis (at least one of the variables used in the comparison was a constant since all participants scored the same) and introduce some bias to the results.

### 3.4. Bland–Altman Analysis

The agreement between the two methods was also analyzed using the Bland–Altman method. Data of the mean difference (bias) and LOA (with their corresponding 95% CI) as well as fitted lineal regression are listed in Table 4, and graphical representation of this analysis is shown in Figure 1. These results confirm an overall overestimation bias of the FFQ-MEDAS against the 3d-FD of ~0.8 units for the examined entire population. At the individual country level, the average bias between the two scores was also ~0.8 units for CY and much smaller and close to 0.0 for GR, BG, and NMK. On the other hand, the estimated bias was close to 1.0 for IT and PT and displayed the highest difference in the Spanish population sample. The fitted linear regression equations were all not significant indicating that there was no evidence of proportional bias.

## 4. Discussion

Questionnaires measuring and scoring adherence to the MD are considered suitable tools for identifying the dietary habits of a given population and for analyzing their association with the potential health benefits of this diet [5,30]. Amongst the various scores that have been developed to measure MD adherence, the 14-item MEDAS screening tool was previously validated for the Spanish population in the PREDIMED study [6] and has been widely investigated for its applicability in several other countries around the world [7,8,9,10]. However, a simultaneous and global validation of the 14-item MEDAS scoring system in countries around the European Mediterranean region had not been previously undertaken. In the present study, we provide for the first time, comprehensive validation data of the 14-MEDAS in several Southern European Mediterranean countries, i.e., GR, PT, IT, SP, and CY as well as in two Balkan bordering countries, NMK and BG.

One of the limitations attributed to the 14-MEDAS was that this score was validated for a specific sample population (i.e., 55–80 years old at a high risk for coronary heart disease), and thus, the results could not be extrapolated to the general population. Moreover, the score was compared to a long (137-item) FFQ as the reference method [6]. Both, FFQ and food diaries have been previously employed as validation reference methods of the 14-MEDAS in different countries [6,7,8,9,10,11,12,13,14]. Although food records are not free from errors, they have been recommended for validating food-frequency questionnaires [20] since they minimize errors related to recall and perception of portion sizes, and also, the measurement errors of the food records do not correlate with those of the 14-MEDAS [9,20,31]. In this study, we have analyzed the validity of the 14-item MEDAS in a general adult population (>18 years old) from different Southern European countries (no other restriction criteria were applied) using a 3d-FD record as a reference method. Since there is not a gold standard unique protocol, we have implemented, globally and in each separate country, a number of analyses recommended to quantify the agreement between methods: (1) test–retest reliability [32], (2) relative and absolute comparison using Pearson correlation, ICC, and food items comparison by κ statistics analysis, and (3) Bland–Altman graphical analysis of the difference between methods [19,20].

Overall, the reliability of our FFQ-MEDAS was good to very good both in the global population as well as in each separate country reinforcing the good stability of the questionnaire over time [9]. Regarding validity and based on the Pearson and ICC correlations, the agreement between the FFQ-MEDAS and the reference 3d-FD was in general categorized as moderate but with individual countries like GR, which reached excellent agreement while BG and NMK displayed a lower concordance. Compared with the validation analysis of the 14-MEDAS in Spain [6], and though the Spanish sample tested in our study (*N* = 40) was much smaller than that used in the PREDIMED study (*N* = 7146), the correlation results were very similar (*r* = 0.50 and ICC = 0.44 in our study vs. *r* = 0.52 and ICC = 0.51 in the PREDIMED study [6]). Likewise, the validation results attained in our Portuguese sample population (*r* = 0.597 and ICC = 0.693) were also comparable to those recently published in a larger Portuguese population of mixed adults (*r* = 0.641 and ICC = 0.634) [14]. The validation of the 14-MEDAS against the 3d-FD in a non-Mediterranean country (UK) also showed a similar moderate agreement (*r* = 0.50, ICC = 0.53) leading those authors to suggest that the potential of this tool to measure MD adherence was similar in Mediterranean and non-Mediterranean countries [9]. In contrast, in the non-Mediterranean countries examined by us, NMK and BG, correlation was poorer than in the Mediterranean participants supporting the need and relevance of further exploring and validating dietary questionnaires in each different country or community investigated.

To further investigate whether any of the differences found in the validation in each separate country might be related to differences in the validation at the food item level, we performed κ analyses of each of the components of the questionnaire. The results supported a general fair to moderate concordance for many of the food items principally in GR, PT, IT, and SP as well as in the entire sample population examined. On the other hand, poorer κ values were found in CY and NMK. In agreement with κ statistics reported in previous similar validation studies [6,7,9,13], we observed that, in general, and with independence of the country and sample population examined, the fair to moderate levels of agreement prevail for most items, and only a few food items reach higher concordance. A closer and more specific comparison between the same countries, i.e., the PREDIMED study [6] conducted in Spain and our Spanish sample population or between the analysis carried out by Gregório et al. [14] and our Portuguese sample showed equivalent results for only three items, i.e., butter, sweet drinks and “sofrito” in the Spanish example and olive oil quantity, vegetables, and butter in the Portuguese case. As indicated by other researchers in the area, different factors can influence these results such as the responders’ ability to remember, describe, and quantify food consumption [14]. Even when the same individuals responded to the same questionnaires within a 3-month difference, the κ analysis per food item showed substantial variability [7]. The κ results are also biased by food items where the total sample population investigated gave the same response (i.e., it is not a variable), and thus, it is not susceptible for the analysis. This was the case of, for example, using olive oil to cook in the sample populations from SP, IT, and GR, where all respondents scored 1 (normally use olive oil to cook). This situation was also particularly apparent in the case of the sample population from BG with eight food items where the score was the same for all participants. Further studies with larger sample population may mitigate this issue.

We next applied the Bland–Altman graphical analysis which is considered one of the most suitable ways to compare methods and that allows for quantification of the concordance between mean differences, as well as concordance of the variability in all individuals (95% agreement intervals) [27,33]. Our analyses were indicative of a general overestimation bias of the MD adherence by the 14-item FFQ-MEDAS questionnaire as compared with the 3d-FD. Nevertheless, there was no trend in the difference between methods as the average increased (absence of systematic bias), and the scatter of values around the bias line was consistent (Figure 1). These results are in good agreement with previous reports where it has been shown that the 14-MEDAS yielded a higher score than other FFQ or food records both in European [6,7,9] and non-European countries [10,11]. Furthermore, the size of the overestimation was highly variable with bias between methods ranging from a minimum value of 0.1 unit in NMK to almost 2.0 units in SP. Similar variability in the mean differences between methods has also been reported in previous studies, with values < 0.5 units as in the PREDIMED validation conducted in Spain (0.25 units) [6] or in an adapted version of the 14-MEDAS in Korea (0.3 units) [10] and cases in which the difference was > 1.0 unit, as in the validation of the 14-MEDAS in the UK (1.5 units) [9] or in Germany (1.3 units) [7]. Our results also show that with independence of the country and sample size, most individual data lay within the estimated LOAs. However, and as in previous published similar analyses [7,9,10], these LOAs cover a rather broad range of differences between methods (~between 4 and 8 units, including negative and positive values). As it has been recently reviewed about the use and reporting of Bland–Altman analyses [28], key information is usually missing in many of those studies: (i) how big the average discrepancy between methods (bias) is acceptable, and (ii) how wide the LOAs can be so that the results are not too ambiguous. These two key points have not yet been addressed within the context of MD adherence measurement. Based on previous analyses [6,7,9,10] and our own results in the Southern European sample population examined, we suggest that a mean acceptable bias must be at least ≤1.0 units for a more accurate qualification of the MD adherence, but future research in the area should continue contributing to delineate this minimum size. It is also essential that acceptable LOAs between the 14-MEDAS and reference methods are established and that we continue investigating the factors that may influence these values.

Overall, and taking into consideration the results of all the analyses implemented, a global moderate validation for the 14-item MEDAS against the 3d-FD was estimated for the complete Southern European population examined. Regarding the individual countries, it appears that the validation was slightly better in the Mediterranean countries but establishing a clear ranking order in the validation performance was, however, not trivial. On the whole, the sample population from GR clearly displayed the best validation scores. The rest of the Mediterranean countries PT, IT, CY, and SP had all moderate levels of agreement but exhibited higher bias values. On the other hand, NMK and BG, which had the lowest correlation values, performed better in the Bland–Altman analysis with the lowest bias values between the two methods. These results confirm differences in the validation performance between countries and reinforce the need to establish improved standardized protocols and qualification criteria to better differentiate MD adherence between countries.

With regards to the estimated 14-item MEDAS scores, in this exploratory study, we found a moderate MD adherence (6.22 ± 2.01) in the whole investigated group of Southern European Mediterranean countries and were ranked from highest to lowest score as follows: SP >> IT > PT > CY ≅ GR >> NMK > BG. These results show that despite the differences in the validation performance, the 14-MEDAS had some capacity to differentiate between a weaker adherence for the Balkan countries against a fair-to-moderate adherence for the Mediterranean representatives. These results are not too dissimilar from those previously reported using another Mediterranean score, the Mediterranean adequacy index (MAI). These countries (except NMK, no information available) were ranked from best to worst MAI as follows: GR > IT > SP > PT > BG > CY. They were also reported to have reduced their MAI scores in the last decades, indicating a shift away from the traditional MD pattern [34]. Further, the sample population from SP, which exhibited the highest adherence to the MD, reached an average MEDAS of 8.35 ± 1.65 and was very close to the value attained in the PREDIMED validation study (8.68 ± 1.90) [6]. Moreover, our estimated IT, PT, and GR 14-MEDAS scores (6.86 ± 1.63, 6.55 ± 1.98 and 6.32 ± 1.68, respectively) were not too dissimilar from the previously published scores of 7.34 ± 1.9 in Southern Italy [35], 7.29 ± 2.15 in a population of Portuguese adults from Lisbon [14], and 6.40 ± 1.90 in a sample population of Greek students [36] and were all classified in the fair-to-moderate adherence range [22]. Conversely, in non-Mediterranean European countries, the 14-item MEDAS scores were lower such as in the UK study with a value of 5.47 ± 2.09 [9]. Overall, these data show a reasonable agreement with previously published Mediterranean scores and support some validity of this instrument to categorize MD adherence in different populations.

It has been already indicated that factors like age, ethnic group, gender, and (or) health status of a population can affect the outcome of the validation of dietary assessment [20,37]. Misreporting is a common reason for the differences found between dietary assessment instruments and can be influenced by factors such as sex [38] or BMI, with the probability of underreporting increasing with a higher BMI [39]. In the same way, the 14-MEDAS score has also been reported to be influenced by sex [36] or age [35] as well as having a negative association with BMI, i.e., higher 14-MEDAS score is correlated with lower BMI [22]. Our analyses further corroborate this negative relationship between BMI and the 14-MEDAS score as well as between BMI and the validation of the 14-MEDAS score against the 3d-FD score, which may partially contribute to explain the results attained in NMK and BG both countries with a significant higher percentage of participants with overweight and obesity than in the other countries. With the increasing numbers of overweight people in Europe, it is of utmost importance that we carefully scrutinize the development, modification, and application of diet adherence measurement tools as well as the factors influencing their implementation.

The main strength of our study is that this is the first time that the 14-item MEDAS has been simultaneously validated in a number of different countries around the Southern European Mediterranean area using the same methodology (a range of well-recommended analyses), and thus, comparisons between them were feasible. The results confirm a fair-to-moderate validation of this tool in the general adult population from those countries and supports its potential applicability to comparatively differentiate MD adherence between Mediterranean and non-Mediterranean European countries; however, an accurate and complete ranking of the countries was not accomplished. Our study has, nonetheless, a number of limitations, i.e., the participation of more volunteers with overweight and obesity in some of the countries, certain difficulty in reporting some foods and portion sizes by the participants, and the limited sample size achieved in some of the investigated countries. In this exploratory analysis, we have preliminarily addressed the BMI influence on the validation of the 14-MEDAS, but more studies in larger populations are needed to further (1) confirm and improve the suitability of the 14-MEDAS tool in these and other countries of the Mediterranean area and of Europe as a whole, (2) optimize the accuracy of the self-report instruments by increasing our knowledge of the interindividual variability in underreporting and of the factors that influence this variability, e.g., differences between types of foods or the individual’s characteristics that affect the responses to dietary questionnaires such as BMI or health status [40] and (3) address the need to adjust in MD studies the effects of specific dietary choices (i.e., vegetarian/vegan subpopulations) [41] or those of alcohol consumption (alcoholic drinkers vs. non-alcoholic drinkers) [42]. Overall, and since the assessment of dietary habits remains essential to advance in the understanding of the diet-health binomial, it is important that we further elaborate and improve standard validation protocols with specific application to measure MD adherence and with a focus on establishing common criteria to optimize the acceptance or rejection of the validation so that more meaningful and accurate comparison between different sample populations becomes possible.

## Figures and Tables

**Figure 1 nutrients-12-02960-f001:**
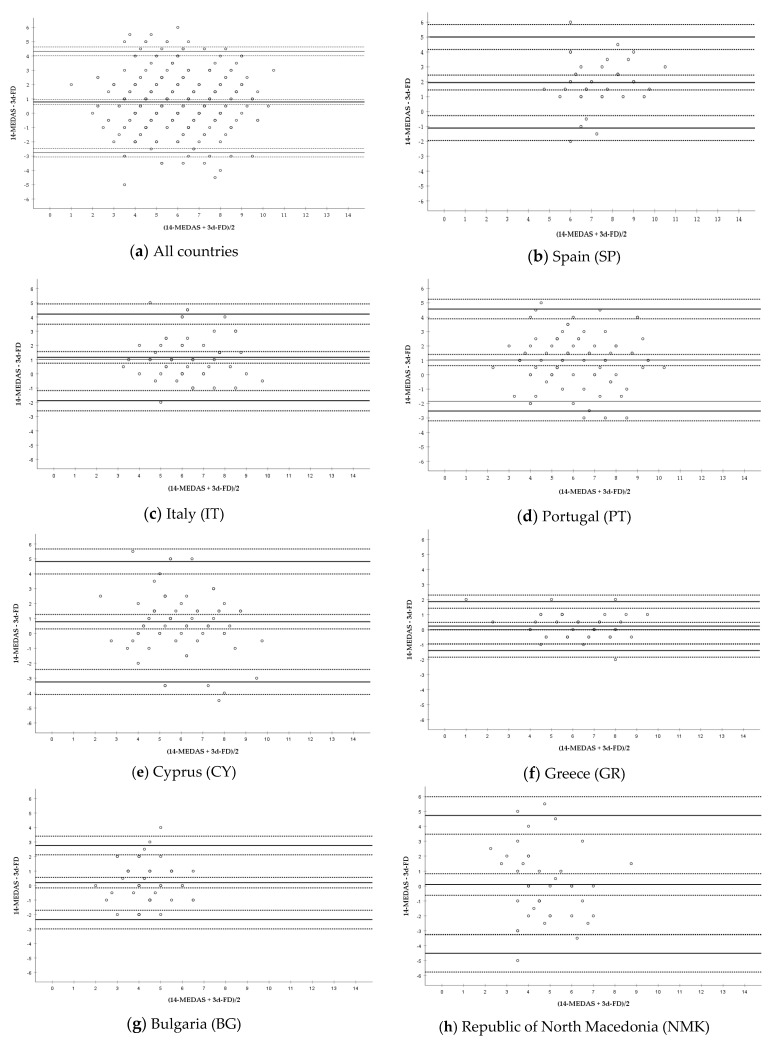
Bland–Altman plots of the differences between the results of the 14-MEDAS questionnaire and those of the 3d-FD. Y axis: DIFFERENCE (14-MEDAS—3d-FD) vs. X axis: MEAN ((14-MEDAS + 3d-FD)/2). Results are presented for: (**a**) All countries together, (**b**) Spain (SP), (**c**) Italy (IT), (**d**) Portugal (PT), (**e**) Cyprus (CY), (**f**) Greece (GR), (**g**) Bulgaria (BG), (**h**) Republic of North Macedonia (NMK).

**Table 1 nutrients-12-02960-t001:** Characteristics and distribution of the sample population.

	All Countries	SP	IT	PT	CY	GR	NMK	BG
*N* (%)	402	40 (10.0)	58 (14.4)	86 (21.4)	72 (17.9)	44 (10.9)	43 (10.7)	59 (14.7)
Sex distribution:								
Women; *N* (%)	238 (59.2)	23 (57.5)	34 (58.6)	57 (66.3)	38 (52.8)	25 (56.8)	25 (58.1)	36 (61.0)
Men; *N* (%)	164 (40.8)	17 (42.5)	24 (41.4)	29 (33.7)	34 (47.2)	19 (43.2)	18 (41.9)	23 (39.0)
Age range (years)	18–81	24–71	19–65	19–73	20–74	18–80	20–81	20–80
Age mean ± SD (years)	39.4 ±15.2	45.9 ± 11	36.1 ±13.5	34.2 ±14.1	35.0 ± 15	34.3 ±12.0	46.3 ±16.2	49.9 ±14.6
Age distribution *N* (%)								
18–24 (years)	85 (21.1)	1 (2.5)	18 (31.0)	29 (33.7)	28 (38.9)	6 (13.6)	2 (4.7)	1 (1.7)
25–34 (years)	104 (25.9)	6 (15.0)	12 (20.7)	26 (30.2)	14 (19.4)	26 (59.1)	11 (25.6)	9 (15.3)
35–44 (years)	75 (18.7)	14 (35.0)	8 (13.8)	8 (9.3)	14 (19.4)	6 (13.6)	10 (23.3)	15 (25.4)
45–54 (years)	58 (14.4)	9 (22.5)	14 (24.1)	12 (20.7)	7 (9.7)	3 (5.2)	4 (9.3)	9 (15.3)
55–64 (years)	47 (11.7)	8 (20.0)	5 (8.6)	7 (8.1)	2 (2.8)	1 (2.3)	10 (23.3)	14 (23.7)
≥65 (years)	33 (8.2)	2 (5.0)	1 (1.7)	4 (4.7)	7 (9.7)	2 (4.5)	6 (14.0)	11 (18.8)
BMI (kg/m^2^)								
mean ± SD	25.2 ± 5.0	23.4 ± 2.6	23.3 ± 3.1	24.0 ± 3.6	24.9 ± 4.7	25.9 ± 5.6	26.9 ± 4.9	28.5 ± 7.3
BMI distribution ^1^ *N* (%)								
Underweight	9 (2.2)	1 (2.5)	1 (1.7)	1 (1.2)	1 (1.4)	0 (0.0)	1 (2.3)	2 (3.4)
Normal weight	223 (55.5)	27 (67.5)	44 (75.9)	54 (63.5)	42 (58.3)	23 (52.3)	15 (34.9)	20 (33.9)
Overweight	113 (28.1)	11 (27.5)	12 (20.7)	24 (28.2)	20 (27.8)	13 (29.6)	17 (39.5)	15 (25.4)
Obesity	57 (14.2)	1 (2.5)	1 (1.7)	6 (7.1)	9 (12.5)	8 (18.2)	10 (23.3)	22 (37.3)
Weight excess (overweight + obesity)	170 (42.3)	12 (30.0)	13 (22.4)	30 (35.3)	29 (40.3)	21 (47.8)	27 (62.8)	37 (62.7)

^1^: Categories of BMI according to the World Health Organization (WHO) [29]: underweight < 18.5 kg/m^2^; normal weight ≥ 18.5–24.9 kg/m^2^; overweight 25.0–29.9 kg/m^2^; obesity ≥ 30.0 kg/m^2^. SP: Spain; IT: Italy; PT: Portugal; CY: Cyprus; GR: Greece; NMK: Republic of North Macedonia; BG: Bulgaria; *N*: sample size; SD: Standard Deviation; BMI: Body Mass Index.

**Table 2 nutrients-12-02960-t002:** Global and single country 14-item Mediterranean Diet Adherence Screener (14-MEDAS) vs. 3-day food diary (3d-FD) scores (mean ± SD). Results of the validation analyses for test reliability, association, and agreement.

*N* (Valid Population) ^1^	FFQ-MEDAS ^2^ (1) FFQ-MEDAS (2)	Test–Retest Reliability ^3^ (*r*, Sig. Bilateral)	FFQ-MEDAS (Mean Score)	3d-FD Score	Correlation ^4^ (*r*, Sig. Bilateral)	ICC ^5^ (95%CI, Sig. Bilateral)
All countries (402)	(1) 6.22 ± 2.03 (2) 6.21 ± 2.14	0.852, <0.001 Strong positive correlation	6.22 ± 2.01	5.43 ± 1.89	0.573, <0.001 Moderate positive correlation	0.692 (0.552, 0.780; <0.001) Moderate
SP (40)	(1) 8.15 ± 1.73 (2) 8.55 ± 1.71	0.837, <0.001 Strong positive correlation	8.35 ± 1.65	6.40 ± 1.46	0.503, 0.001 Moderate positive correlation	0.440 (−0.210, 0.745; <0.001) Poor
IT (58)	(1) 6.90 ± 1.68 (2) 6.83 ± 1.74	0.809, <0.001 Strong positive correlation	6.86 ± 1.63	5.71 ± 1.63	0.546, <0.001 Moderate positive correlation	0.610 (0.150, 0.802; <0.001) Moderate
PT (86)	(1) 6.54 ± 2.04 (2) 6.55 ± 2.10	0.827, <0.001 Strong positive correlation	6.55 ± 1.98	5.52 ± 2.02	0.597, <0.001 Moderate positive correlation	0.693 (0.420, 0.824; <0.001) Moderate
CY (72)	(1) 6.33 ± 1.90 (2) 6.32 ± 2.03	0.623, <0.001 Moderate positive correlation	6.33 ± 1.77	5.54 ± 2.06	0.427, <0.001 Moderate positive correlation	0.564 (0.299, 0.728; <0.001) Moderate
GR (44)	(1) 6.41 ± 1.67 (2) 6.23 ± 1.83	0.842, <0.001 Strong positive correlation	6.32 ± 1.68	6.09 ± 1.87	0.895, <0.001 Strong positive correlation	0.939 (0.887, 0.967: <0.001) Excellent
NMK (43)	(1) 4.93 ± 1.62 (2) 4.67 ± 1.76	0.919, <0.001 Strong positive correlation	4.80 ± 1.66	4.70 ± 1.91	0.131, 0.401 No correlation	0.234 (−0.434, 0.588; 0.200) No agreement
BG (59)	(1) 4.46 ± 1.21 (2) 4.49 ± 1.21	0.930, <0.001 Strong positive correlation	4.47 ± 1.19	4.27 ± 1.19	0.311, 0.016 Weak correlation	0.473 (0.118, 0.686; 0.008) Poor

^1^*N* (valid population used in the analyses). ^2^ FFQ-MEDAS: Food frequency questionnaire designed to measure the 14-MEDAS score. ^3^ Pearson correlation between FFQ-MEDAS (1) and FFQ-MEDAS (2). ^4^ Pearson correlation between 3d-FD and FFQ-MEDAS (mean). ^5^ ICC: Intraclass Correlation Coefficient between 3d-FD and 14-MEDAS (mean) using the two-way mixed model and absolute agreement. Bilateral significance considered for *p*-value < 0.05.

**Table 3 nutrients-12-02960-t003:** Relative agreement (food frequency questionnaire (FFQ-MEDAS) vs. 3d-FD): per-item validation κappa statistics analysis and level of agreement in all the countries.

Question (Score)	All Countries	SP	IT	PT	CY	GR	NMK	BG
1. Olive oil (yes)	0.590 Moderate	NA ^1^	NA	0.133 Slight	−0.003 No agreement	NA	0.225 Fair	0.871 Almost perfect
2. Olive oil (≥4)	0.361 Fair	0.228 Fair	−0.063 No agreement	0.390 Fair	−0.084 No agreement	0.488 Moderate	NA	−0.017 No agreement
3. Vegetables (≥2)	0.184 Slight	0.000 No agreement	0.419 Moderate	0.252 Fair	0.222 Fair	0.485 Moderate	0.166 Slight	NA
4. Fruits (≥3)	0.502 Moderate	0.459 Moderate	0.181 Slight	0.549 Moderate	0.391 Fair	0.560 Moderate	−0.042 No agreement	NA
5. Red meat (<1)	0.114 Slight	−0.080 No agreement	0.110 Slight	−0.228 Disagreement	NA	0.440 Moderate	0.557 Moderate	NA
6. Butter (<1)	0.257 Fair	0.655 Substantial	0.270 Fair	0.124 Slight	0.030 Slight	0.455 Moderate	−0.307 Disagreement	0.168 Slight
7. Sweet drinks (<1)	0.281 Fair	0.362 Fair	0.097 Slight	0.449 Moderate	0.003 No agreement	0.307 Fair	0.125 Slight	0.140 Slight
8. Wine (7 to 14)	0.391 Fair	0.538 Moderate	0.545 Moderate	0.223 Fair	NA	0.116 Slight	0.482 Moderate	0.676 Substantial
9. Legumes (≥3)	0.264 Fair	0.275 Fair	0.467 Moderate	0.124 Slight	0.126 Slight	0.540 Moderate	−0.116 Disagreement	NA
10. Fish (≥3)	0.239 Fair	0.366 Fair	0.098 Slight	0.126 Slight	0.099 Slight	0.340 Fair	−0.040 No agreement	NA
11. Desserts (<3)	0.333 Fair	0.498 Moderate	0.446 Moderate	0.268 Fair	0.035 Slight	0.035 Slight	0.094 Slight	NA
12. Nuts (≥3)	0.403 Fair to moderate	0.659 Substantial	0.268 Fair	0.361 Fair	0.300 Fair	0.836 Almost perfect	0.055 Slight	NA
13. White meat (≤1 or yes)	0.234 Fair	0.050 Slight	0.242 Fair	0.298 Fair	0.222 Fair	0.690 Substantial	0.073 Slight	0.050 Slight
14. ‘Sofrito’ (≥2)	0.204 Slight to fair	0.050 Slight	0.190 Slight	−0.024 No agreement	0.062 Slight	0.919 Almost perfect	0.206 Fair	NA

^1^: NA, not applicable (at least one of the variables used in the comparison was a constant, i.e., the same score was obtained for all respondents). Values corresponding to fair levels of agreement or above are shaded in grey color.

**Table 4 nutrients-12-02960-t004:** Bland–Altman Analysis of the FFQ-MEDAS vs. 3d-FD.

Country (*N*) ^1^		Bland–Altman Analysis		
	Mean Difference ^2^ (Bias) ± SD (95% CI)	Upper LOA (95% CI)	Lower LOA (95% CI)	Fitted Linear Regression (Sig. Bilateral)
All countries (402)	0.79 ± 1.81 (0.61, 0.96)	4.33 (4.02, 4.63)	−2.75 (−3.06, −2.45)	y = 0.35 + 0.08x (0.150)
SP (40)	1.95 ± 1.56 (1.45, 2.45)	5.01 (4.17, 5.84)	−1.11 (−1.94, −0.27)	y = 0.78 + 0.16x (0·399)
IT (58)	1.16 ± 1.55 (0.75, 1.56)	4.20 (3.49, 4.91)	−1.89 (−2.60, −1.18)	y = 1.19 − 0.01x (0·974)
PT (86)	1.02 ± 1.81 (0.64, 1.41)	4.57 (3.89, 5.25)	−2.52 (−3.20, −1.84)	y = 1.79 − 0.03x (0.815)
CY (72)	0.78 ± 2.06 (0.30, 1.27)	4.82 (3.99, 5.66)	−3.25 (−4.09, −2.41)	y = 2.03 − 0.21x (0·167)
GR (44)	0.23 ± 0.83 (−0.03, 0.48)	1.86 (1.42, 2.30)	−1.40 (−1.84, −0.96)	y = 0.91 − 0.11x (0·137)
BG (59)	0.20 ± 1.39 (−0.16, 0.57)	2.76 (2.11, 3.40)	−2.35 (−3.00, −1.71)	y = 0.20 + 0.001x (0·998)
NMK (43)	0.10 ± 2.36 (−0.62, 0.83)	4.72 (3.47, 5.98)	−4.51 (−5.77, −3.26)	y = 1.30 − 0.25x (0·361)

^1^*N* (valid population used in the analyses). ^2^ Difference: FFQ-MEDAS score—3dFD score. LOA: Limits of agreement.

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
