# Peer review of "Exploring the Validity of the 14-Item Mediterranean Diet Adherence Screener (MEDAS): A Cross-National Study in Seven European Countries around the Mediterranean Region"

_nutrients, 2020, doi:10.3390/nu12102960_

Round 1

Reviewer 1 Report

This work collects the first study of 14-MEDAC simultaneously in different Mediterranean and non-Mediterranean countries. A diverse population of each country was selected for performing the study, not taking their sex, age or physical condition into account.

Minor changes

Line 101: a dot is missing after references 17 and 18.

Line 171: some decimals are represented by the wrong punctuation mark.

FFQ questionnaire (Suplementary Table 1): I have two considerations to make about the questions included.

  1. First one, I would like to remark that some people refuse to drink alcohol for health reasons so that wine is not part of their diet. This fact would correspond to a lower MD adherence which could be disguised as an unhealthy habit, which is not necessarily true. I would like the authors to take this into consideration in the text.
  2. Secondly, although statistically most likely this was not given between the participants, there is a possibility that some of them were vegetarian/vegan, in which cases meat or fish are not included in their diet for different reasons than health. I appreciate that this is taken into account in question 13 about chicken/rabbit/turkey meat, where the authors offer the alternative of vegetable protein sources, but this consideration is not made in question 10 about fish/shellfish. I think that the authors should include in the text that in next studies the alternative of nuts and chia/flax seeds for replacing omega-3 in fish should also be considered. It also should be included in the text whether some of the participants were vegetarian/vegan or not, and/or if the authors preferred to not include these populations in the study in order to maintain a classical MD pattern.

Author Response

We thank the reviewer for her/his kind comments on our manuscript. We have corrected the punctuation marks as indicated and added a sentence/references in the Discussion regarding the important issue of specific consumers such as vegetarian/vegan individuals and (or) the consumption of wine or other alcoholic drinks and their relation to health. Indeed, there are still many important issues to solve in the area of diet assessment. We are currently working on the data from a larger study where we hope to be able to further address some of these aspects.  

Reviewer 2 Report

Mediterranean Diet (MD) is characterized by the daily consumption of olive oil, whole grains, fruits, vegetables, nuts, fish, as well as a moderate intake of lean fresh meat, and dairy products. This diet is considered as one of the healthiest dietary patterns as previous studies showed that MD reduced the risk of overall mortality, cardiovascular diseases, cancer incidence, neurodegenerative diseases, and type-II diabetes.

The 14-item MEDAS questionnaire was indicated to be a moderate and reasonably valid tool for the rapid estimation of MD adherence. However, a simultaneous and comparative cross-national validation of the 14-MEDAS tool in different countries of the Mediterranean region has never been undertaken. The study aimed to explore the applicability of a 14-item MEDAS questionnaire to the general adult population from several Southern European countries around the Mediterranean area with different dietary cultures.

They confirmed a fair-to-moderate validation of the 14-MEDAS tool in the general adult population and supports its potential applicability to comparatively differentiate MD adherence between the Mediterranean and non-Mediterranean European countries.

The findings in this manuscript were innovative and offered new insights on the validation of the 14-MEDAS, although this is not a large-scale study. This work is well organized and comprehensively described. I suggest you shorten the Discussion section a little bit.

Author Response

We thank the reviewer for her/his kind comments on our manuscript. Regarding the length of the Discussion, we hope that this is not a major problem for the publication of the article. We considered of interest to address several important issues and thus we have not been able to shorten it. Indeed, we have added a new short sentence to address some additional comments from the other reviewer. This shows interest in the subject and the need to continue working on the improvement of diet assessment tools.